# The Distal Promoter of the *B438L* Gene of African Swine Fever Virus Is Responsible for the Transcription of the Alternatively Spliced *B169L*

**DOI:** 10.3390/v16071058

**Published:** 2024-06-30

**Authors:** Hongwei Cao, Hao Deng, Yanjin Wang, Diqiu Liu, Lianfeng Li, Meilin Li, Dingkun Peng, Jingwen Dai, Jiaqi Li, Huaji Qiu, Su Li

**Affiliations:** State Key Laboratory for Animal Disease Prevention and Control, National African Swine Fever Para-Reference Laboratory, National High-Containment Facilities for Animal Disease Control and Prevention, Harbin Veterinary Research Institute, Chinese Academy of Agricultural Sciences, Harbin 150069, China; chw18344306620@163.com (H.C.); denghao333417@163.com (H.D.); wangyanjin1996@126.com (Y.W.); liudiqiu@caas.cn (D.L.); lilianfeng@caas.cn (L.L.); m18037901611@163.com (M.L.); pengdk11@163.com (D.P.); dai_wener@163.com (J.D.); lijiaqi1051951442@163.com (J.L.)

**Keywords:** African swine fever virus, *B438L* promoter, transcription initiation, *B169L* protein

## Abstract

The B169L protein (pB169L) of African swine fever virus (ASFV) is a structural protein with an unidentified function during the virus replication. The sequences of the *B169L* gene and the downstream *B438L* gene are separated by short intergenic regions. However, the regulatory mode of the gene transcription remains unknown. Here, we identified two distinct promoter regions and two transcription start sites (TSSs) located upstream of the open reading frame (ORF) of *B438L*. Using the promoter reporter system, we demonstrated that the cis activity of the ORF proximal promoter exhibited significantly higher levels compared with that of the distal promoter located in the *B169L* gene. Furthermore, transfection with the plasmids with two different promoters for *B438L* could initiate the transcription and expression of the *B438L* gene in HEK293T cells, and the cis activity of the ORF proximal promoter also displayed higher activities compared with the distal promoter. Interestingly, the *B438L* distal promoter also initiated the transcription of the alternatively spliced *B169L* mRNA (*B169L* mRNA2) encoding a truncated pB169L (tpB169L) (amino acids 92–169), and the gene transcription efficiency was increased upon mutation of the initiation codon located upstream of the alternatively spliced *B169L* gene. Taken together, we demonstrated that the distal promoter of *B438L* gene initiates the transcription of both the *B438L* mRNA and *B169L* mRNA2. Comprehensive analysis of the transcriptional regulatory mode of the *B438L* gene is beneficial for the understanding of the association of B438L protein and pB169L and the construction of the gene-deleted ASFV.

## 1. Introduction

African swine fever (ASF), caused by African swine fever virus (ASFV), is a highly contagious and hemorrhagic disease of swine, with a high mortality of up to 100% [1]. Since its initial outbreak in Kenya in the 1920s, ASF predominantly manifested as an epidemic in Europe, South America, and the Caribbean [2]. However, the disease was introduced into China in 2018 and resulted in huge economic losses due to the lack of effective vaccines and antiviral drugs [3,4]. ASFV belongs to the family *Asfarviridae* and is a large enveloped virus with a genome of double-stranded DNA, encoding more than 50 structural proteins and 100 nonstructural proteins, many of which have undetermined functions [5]. It is an important strategy to investigate the functions of viral proteins through constructing a gene-deleted ASFV. The identification of viral gene promoters will be beneficial for the construction of the gene-deleted ASFV mutants. By scanning the complete genome of ASFV, we demonstrated that the viral structural genes *B438L* and *B169L* are densely distributed in an organized manner, implying the transcriptional association between the viral genes.

The *B169L* gene encodes the B169L protein (pB169L) of 169 amino acids (aa). It is noteworthy that the transcription of *B169L* gives rise to two mRNAs: the full-length *B169L* mRNA and the alternatively spliced *B169L* mRNA (referred to as *B169L* mRNA2) encoding a truncated pB169L (tpB169L) (aa 92–169) [6]. However, the transcription start site (TSS) and the promoter that are responsible for the transcription of the *B169L* mRNA2 remain unknown. Notably, the *B169L* gene is arranged adjacent with its downstream gene *B438L* in the ASFV genome. Therefore, we postulated that the promoter sequence of the downstream gene *B438L* is located within the *B169L* gene.

A previous study has revealed that the ASFV B438L protein (pB438L) is indispensable for the construction of the icosahedral capsid of ASFV, and the gene-deletion of *B438L* leads to a significant alteration in virion assembly, resulting in the formation of tubular viral particles instead of the characteristic icosahedral capsid, thereby disrupting the icosahedral symmetry [7,8]. The morphological perturbation also affects the infectivity and pathogenicity of viruses. It has been shown that the transcriptional machinery of ASFV exhibits complexity, which is involved in the complex interplay between enzymes and cofactors encoded by ASFV [9]. The ASFV *B438L* and *B169L* ORFs are densely distributed, it is possible that the upstream gene encompasses promoter(s) to govern the expression of the downstream gene(s) of ASFV, and an individual gene may harbor multiple promoters for the initiation of the transcription of downstream viral genes. Therefore, it is essential to decipher the transcriptional regulation patterns of the downstream *B438L* gene, which will facilitate the design of the *B169L*-deleted ASFV vaccines.

In this study, we elucidated an intricate interrelationship between the ASFV *B438L* gene and its adjacent upstream and downstream genes. Notably, the *B438L* promoter region is located in the upstream of its ORF and encompasses two distinct promoters, and the *B438L* distal promoter also initiates the transcription of the alternatively spliced *B169L* (*B169L*-2). Moreover, we have discerned that the transcriptional activity of the *B438L* gene depends on the viral transcriptional machinery, in conjunction with relevant cofactors, and the transcription efficiency of the gene was also affected by the translation initiation codon (ATG) located upstream of *B169L-2*. The findings enhance our understanding of the transcriptional mechanisms of ASFV, thereby providing novel insights into the development of antiviral strategies.

## 2. Materials and Methods

### 2.1. Cells and Viruses

Primary porcine alveolar macrophages (PAMs) were collected from the alveolar lavage fluid of 20-day-old specific-pathogen-free (SPF) pigs. The PAMs or HEK293T (ATCC) cells were cultured in RPMI 1640 (catalog no. C11875500BT; Gibco, New York, NY, USA) or Dulbecco’s modified Eagle’s medium (catalog no. D6221; Sigma-Aldrich, Darmstadt, Germany) supplemented with 10% fetal bovine serum (FBS) (catalog no. 10099-141C; Gibco, Thornton, Australia), 100 units/mL penicillin, and 100 μg/mL streptomycin (catalog no. 15240-062; Gibco, Carlsbad, CA, USA) at 37 °C with 5% CO_2_. The ASFV HLJ/2018 strain was isolated from field pig samples in China and propagated in the PAMs as described previously [10].

### 2.2. 5′ and 3′ Rapid Amplification of cDNA Ends (RACE) Assay

The total RNA was extracted from the ASFV-infected PAMs (the target cells of ASFV) at a multiplicity of infection (MOI) of 2 using an RNA purification kit (catalog no. BSC52M1; BioFlux, Hangzhou, China) according to the manufacturer’s instructions. The 5′RACE was performed using the 5′-Full RACE kit (catalog no. 638858; TaKaRa, Mountain View, CA, USA) and the *B438L*-specific primers (B438L-GSP1 and -GSP2 primers). The gene products of the 5′RACE polymerase chain reaction (PCR) were cloned into the pMD19-T vector (catalog no. 639648; TaKaRa, Mountain View, CA, USA) and sequenced using the M13 primers. All the primers in this study are shown in Table 1.

### 2.3. Construction of Plasmids

The genomic DNA of ASFV was extracted using a MagaBio plus virus DNA purification kit (catalog no. 9109; BioFlux, Hangzhou, China) according to the manufacturer’s instructions. Six primer pairs were designed for the amplification of the *B438L-1* to *-6* genes using the genomic DNA as a template. To generate the gene constructs of *B169L-2* and *B169L-mATG-2* (the mutation of the second ATG in *B169L*), a series of reverse primers and intermediate primers were generated and subjected to PCR using the *B438L-1* gene as a template. The mutation and deletion of the GC and TATA boxes in *B438L-7* and *-5* were obtained by sequence overlapping extension (SOE) PCR, and the primers used in this study are shown in Table 1. All the PCR products were purified by using a gel extraction kit (catalog no. 740609.10; TaKaRa, Mountain View, CA, USA) and subsequently digested with the endonucleases *Kpn*I (catalog no. 1618; TaKaRa, Mountain View, CA, USA) and *Xho*I (catalog no. 1635; TaKaRa, Mountain View, CA, USA), followed by a second gel purification for subsequent assays. The sequences of the *B438L* promoters were cloned into the pGL3-Basic luciferase reporter plasmid (catalog no. E1751; Promega, Madison, WI, USA). The recombinant plasmids were subjected to DNA sequencing.

### 2.4. Luciferase Reporter Assay

HEK293T cells cultured in 24-well plates were transfected with 0.5 μg of a series of reporter plasmids harboring different *B438L* promoters, a negative control (pGL3-Basic), and a positive control (pGL3-Control), respectively, together with 0.05 μg of an internal control plasmid pRL-TK expressing the *Renilla* luciferase.

### 2.5. Western Blotting Analysis

The HEK293T cells transfected with the plasmids were lysed with RIPA buffer and subjected to SDS-PAGE analysis, and then, the protein bands were electro-transferred onto nylon membranes. Subsequently, the membranes were blocked in TBS containing 5% skimmed milk for 2 h at room temperature and incubated with a mouse anti-Flag monoclonal antibody (MAb) (catalog no. F7425, Abcam, Cambridge, UK) or mouse anti-*β*-actin polyclonal antibodies (PAb) (catalog no. ZMS1156; Sigma-Aldrich, Darmstadt, Germany) for 2 h. The membranes were rinsed three times using TBS containing 0.05% Tween 20 (TBST) and incubated with IRDye 800CW goat anti-mouse IgG secondary antibody (catalog no. 926-32210; LI-COR, Lincoln, Dearborn, MI, USA) for 1 h at room temperature. Next, the membranes were scanned using an Odyssey infrared imaging system (LiCor BioSciences).

### 2.6. RNA Extraction and RT-qPCR

Total RNA was extracted from the PAMs treated with specific inhibitors or virus infection by a Simply P total RNA extraction kit (catalog no. BSC52M1; BioFlux, Hangzhou, China). The isolated RNA was reverse-transcribed to cDNA using a FastKing gDNA Dispelling RT SuperMix (catalog no. KR118-02; Tiangen, Shanghai, China) according to the manufacturer’s protocols. The cDNAs were used to analyze the mRNA transcription of *B169L*-2 by RT-qPCR using a QuantStudio system (Applied Biosystems), as described previously [11]. The relative mRNA levels of the target genes were normalized to that of the internal reference glyceraldehyde-3-phosphate dehydrogenase (GAPDH), and all primers are listed in Table 1. The relative fold changes in gene expression were determined by the threshold cycle (2^−ΔΔCT^) method [12].

### 2.7. Statistical Analysis

Statistical analysis was performed using GraphPad Prism 8.0 (San Diego, CA, USA). Statistical differences between groups were assessed by Student’s *t* test. All the experiments were performed independently in triplicates. Error bars represent standard deviations (SDs) or standard errors of the mean (SEM) in each group, as indicated in the figure legends (ns, not significant, *p* > 0.05; *, *p* < 0.05; **, *p* < 0.01; ***, *p* < 0.001). A *p*-value of < 0.05 was considered significant.

## 3. Results

### 3.1. Two Distinct Initiation Sites for B438L Transcription

To analyze the TSS located upstream of the *B438L* ATG, 5′RACE assay was conducted using the total RNA of the PAMs infected with the ASFV HLJ/2018 strain at 24 h postinfection (hpi). The first round of amplification was run by using the *B438L*-GSP1 primer pairs. The products of the first round of PCR (5 μL) were used as a template to perform the second round of amplification using the *B438L*-GSP2 primer pairs (Figure 1A). As shown in Figure 1B, PCR products were generated, indicating that the ASFV transcripts were successfully obtained. Approximately 20–25 bacterial clones from each 5′RACE product were sequenced and confirmed to harbor identical 5′-ends, indicating that they were derived from the same *B438L* mRNA transcript. The two positive bands with 700 and 1100 bp, respectively, were mainly amplified by PCR (Figure 1B). Upon sequence alignment, we identified two distinct TSSs located at nt 97563 (TSS-1) and 97848 (TSS-2) of the genome of the ASFV HLJ/2018 strain (Figure 1B). In addition, 3′RACE assay was performed to ascertain the transcription termination site of *B438L*, which was determined to be located at nt 95887 (Figure 1B). Collectively, these findings suggest the presence of two TSSs for the *B438L* gene.

### 3.2. Mapping of the B438L Promoter Region Located in the B169L Gene

To delineate the promoter regions of ASFV *B438L*, a series of primers were designed to amplify seven DNA fragments of the *B438L* gene upstream, using the ASFV genomic DNA as the template (Table 1). Each promoter region was delineated with respect to an individual TSS position, as illustrated in Figure 2A. Notably, these selected promoter regions were designed to lack the ATG codon in the 3′ flanking sequence downstream of the TSS, thereby preventing any potential interference with the translation of the luciferase gene. To evaluate the potential promoter activity of these regions, the DNA fragments were cloned into pGL3-Basic to give rise to the luciferase reporter plasmids.

Subsequently, HEK293T cells were transfected with each of the reporter plasmids of different promoter regions and the internal control plasmid pRL-TK for 24 h. Afterward, the cells were lysed with the passive buffer, and the cell lysates were subjected to the examination of the luciferase activities using the dual-luciferase reporter assay system according to the manufacturer’s instructions. As shown in Figure 2B, none of the reporter plasmids displayed any promoter activities. In light of this observation, we presumed that the transcription of the *B438L* gene is likely to depend on ASFV-specific RNA polymerases and other essential transcription factors. To verify the hypothesis, HEK293T cells were transfected with the *B438L* gene reporter plasmids for 12 h. Subsequently, the cells were infected with an HEK293T cell-adapted ASFV, the ASFV-P61 strain [13], at an MOI of 0.1 for 24 h. Next, the luciferase activities were examined as described above. The results showed that all the reporter plasmids displayed promoter activities, while the *B438L-1*, *B438L-3*, and *B438L-5* exhibited higher activities in comparison with *B438L-2*, *B438L-4*, and *B438L-6* (Figure 2C). More specifically, the promoter activities of *B438L-2* and *B438L-4* were shown to be lower than those of *B438L-3* and *B438L-5*, respectively (Figure 2D and E). To further determine whether a promoter exists between *B438L-3* and *B438L-5*, we constructed a luciferase reporter plasmid in the genome region, and the data showed that *B438L-7* also exhibited higher activity compared with the negative control (Figure 2F). Taken together, these findings suggest that *B438L-5* and *B438L-7* are responsible for initiating *B438L* transcription at TSS-1 and TSS-2, respectively.

### 3.3. Identification of the TATA and GC Boxes in the B438L Promoter

We identified two upstream fragments of the *B438L* TSS as promoters. Utilizing a comprehensive analysis of the locations of TATA or GC box within the promoter regions, along with computational analysis [14,15,16,17], we predicted that the putative TATA and GC boxes are positioned between 10 and 100 nt, respectively, upstream of TSS (Figure 3A). The TSSs of ASFV-*B438L* were determined at nt 97563 and 97848 in the ASFV HLJ/2018 strain.

We predicted the GC and TATA boxes in the two promoters of the *B438L* gene. The primers for amplifying the *B438L* promoter regions are shown in Table 1. The B438L-GSP1 and -GSP2 primers were designed based on the genome of the ASFV HLJ/2018 strain, while the 5′RACE Long and Short Universal Primers were provided by the 5′RACE kit. The TATA box or GC box was predicted as described above.

To identify the GC box or TATA box in the regions of *B438L-7* and *B438L-5*, we introduced mutations or deletions (Figure 3B, upper panel) in these *cis*-acting elements using the mutant primers (Table 1). As shown in Figure 3B, the mutation (*B438L-5*-mGC) or deletion (*B438L-7*-dGC) of the GC box and the mutation of the TATA box (*B438L-5*-mTA) resulted in a reduction in the promoter activity. The data indicate that the *B438L* promoters contain functional TATA and GC boxes, which are indispensable for the promoter activity.

To further validate the identified promoter regions, we generated a series of recombinant plasmids expressing *B438L* using the clonal vector pMD-19T, each of which carries one or two of the two *B438L* promoter regions. Subsequently, HEK293T cells were transfected with the plasmids harboring *B438L* under the control of various regions of the promoters, pMD-Flag-B438L-I (the *B438L-5* region), -II (the *B438L-7* region), and -III (the *B438L-3* region), or the empty pMD-19T vector, respectively. At 36 h post-transfection (hpt), the cell lysates were analyzed by Western blotting to verify the expression of pB438L. Notably, pB438L was efficiently expressed in the cells transfected with different plasmids (Figure 3C), indicating that both the distal and proximal promoters of *B438L* can initiate the expression of pB438L.

### 3.4. Identification of the B438L Distal Promoter That Initiates Transcription of B169L-2

Since the *B438L* distal promoter is located in the *B169L* gene, we investigated whether the promoter affects the *B169L* expression. To this end, we reviewed the relevant data and found that there is *B169L*-mRNA2 encoding a truncated pB169L (tpB169L) (aa 92–169). The sequence of *B169L-2* is located in the downstream of the *B438L* distal promoter. Thus, we verified whether the *B438L* distal promoter is capable of initiating the transcription of *B169L-2*. Firstly, we constructed the recombinant plasmid pMD-Flag-tB169L, which carries the gene sequence of *B169L-2* as well as the distal promoter sequence of *B438L*. Secondly, HEK293T cells were transfected with pMD-Flag-tB169L or pMD-Flag-B438L-II and infected with ASFV-P61. At 24 hpi, the cell lysates were analyzed by Western blotting to verify the expression of tpB169. Notably, tpB169L was expressed in the pMD-Flag-tB169L-transfected HEK293T cells (Figure 4A), indicating that the distal promoter of *B438L* can initiate the expression of tpB169L.

We further analyzed the sequences of *B169L-2* and revealed that the gene contains two ATGs (ATG-2 and ATG-3), located in the region between TSS-2 and the termination codon TAA-1 (Figure 4A). Further investigations demonstrated that there is a termination codon (TAA-2) located in the 15 nt downstream of ATG-2, and ATG-3 is required for the translation of *B169L*-2. We speculated that the transcription efficiency of *B169L-2* was impaired due to the presence of a mini transcript (located in the region of ATG-2 and TAA-2). Thus, we mutated the ATG-2 of *B169L* and constructed the Flag-tagged mutant plasmid pMD-B169L-mATG-2. Next, HEK293T cells were transfected with pMD-tB169L and pMD-B169L-mATG-2 and infected with ASFV-P61. At 24 hpi, cell lysates were collected to examine the protein expression of pMD-Flag-tB169L and pMD-tB169L-mATG2 by Western blotting, and total RNA was extracted for reverse transcription–quantitative PCR (RT-qPCR) to determine the mRNA transcription. The results showed that the transcription and translation efficiency of pMD-tB169L-mATG2 were significantly higher than those of pMD-Flag-tB169L (Figure 4B,C), indicating that the *B438L* distal promoter is capable of initiating the transcription of *B438L* as well as *B169L-2*, and that the mutation of the ATG-2 located between the TSS-2 of *B438L* and the ATG of *B169L-2* (ATG-3) can efficiently enhance the expression of *B169L-2*.

## 4. Discussion

Several studies have focused on the regulation of gene transcription in eukaryotic cells. However, the modulation of viral gene transcription is still insufficient, especially in ASFV, a large DNA virus. The regulatory mechanisms of the genome transcription of ASFV remain largely unknown. The transcription of the ASFV genome exhibits different time courses, which are regulated by early or late promoters [18]. The ASFV p30 (encoded by the *CP204L* gene) and p72 (encoded by the *B646L* gene), two important structural proteins of ASFV, are expressed in the early and late stages, respectively, of the virus life cycle [19,20]. However, the promoters and the transcription time courses of other viral genes remain to be elucidated.

In this study, we revealed that two promoter regions are responsible for the ASFV *B438L* transcription located upstream of the *B438L* ORF. In addition, we demonstrated that the *B169L* gene (the upstream gene of *B438L*) contains a distal promoter for *B438L*. Interestingly, this distal promoter not only initiates the transcription of the *B438L* gene but also governs the transcription of *B169L-2*, suggesting the precise regulatory association of the adjacent viral genes. To further identify the key elements of the promoters, the TATA and GC box regions of the *B438L* promoters were predicted and verified by using site-directed mutagenesis and serial deletions of the boxes. The promoter activity significantly decreased upon mutation of the TATA box. Similarly, mutation or deletion of the GC box also led to a significant reduction in promoter activity. Collectively, we established a novel approach to elucidate the transcriptional association between *B438L* and *B169L,* wherein viral gene promoters were identified through the transfection of a promoter reporter in conjunction with the infection with an HEK293T cell-adapted ASFV mutant.

Furthermore, we have also observed an initiation codon (ATG-2) located upstream of *B169L-2* (Figure 4A). Unexpectedly, the substitution of the ATG with GCT resulted in an enhanced transcription and translation efficiency for *B169L-2* (Figure 4B). The presence of the ATG-2 is likely to be involved in modulating the expression of the downstream *B169L-2*, while the mutation of ATG-2 is speculated to abolish the binding between transcription factors and the promoter, thereby releasing the transcription factors and favoring the downstream gene transcription. The transcription time course of *B169L-2* is also different from that of *B169L*, since the late promoter initiates the *B169L* transcription, while the early promoter initiates the transcription of *B169L-2*. Here, we identified the transcriptional regulatory element of *B169L-2*, indicating that ASFV employs multiple strategies to regulate gene expression. Future studies are required to investigate the functional role of *B169L-2* during ASFV replication and the design of the *B169L* gene-deleted ASFV. The identification of early and late promoters of ASFV will greatly facilitate the investigation into the regulatory mechanisms governing the expression of viral proteins during viral infection.

In addition, in the absence of viral infection, all of the luciferase reporter plasmids were devoid of promoter activity. In contrast, in the presence of viral infection, all of the luciferase reporter plasmids exhibited promoter activities, suggesting that the regulation of the two promoters requires the participation of the ASFV-encoded polymerases and their associated factors (Figure 2B,C). ASFV is a large nucleoplasmic DNA virus that replicates predominantly in the cytoplasm and is involved in the nuclear translocation of the viral genome [21]. However, there is no convincing evidence to support genome replication in the nucleus, which appears to be similar to the members of the *Poxviridae* family that replicate exclusively in the cytoplasm [22]. It is likely that the transcription of several viral genes occurs in the nucleus in the early stage of ASFV replication, and then, the mRNAs are subjected to nuclear export and subsequent translation [23]. The soluble extracts of ASFV particles were found to possess the ability of initiating the transcription system in vitro [24,25], indicating that the ASFV genome encodes multiple viral RNA polymerases and transcription factors.

Viruses have evolved multiple strategies to regulate viral gene expression to complete the virus life cycle, some of which contain multiple promoters to facilitate the selective activation of gene transcription in response to diverse conditions, thereby enhancing viral adaptation in host cells. It has been shown that the *X* gene of hepatitis B virus (HBV), the *nef* gene of HIV-1, and the *UL34* gene (encoding ICP0) of herpes simplex virus 1 (HSV-1) contain multiple promoters, which utilize multiple promoters to regulate gene expression patterns for the adaptation to host diversity, maintaining stable functionality at multiple stages of the virus life cycle [26,27,28,29,30,31]. Moreover, the presence of multiple promoters contributes to the genetic diversity of viral genes, thereby facilitating immune evasion of host cells. The *IE1* gene of human cytomegalovirus (CMV) and the *LMP1* gene of Epstein–Barr virus (EBV) also have multiple promoters, with these promoters playing a pivotal role in viral infection. Different promoters are activated during various stages of the viral life cycle, favoring virus replication and transmission [32,33,34,35]. In this study, we demonstrated that the distal promoter of the *B438L* gene initiates the transcription of both the *B438L* mRNA and *B169L* mRNA2, enhancing the understanding of viral gene transcription. Interestingly, in comparison with the promoters of different ASFV genes, we observed a significant enrichment in A and T in the promoters (Figure 5), indicating that ASFV has evolved unique regulatory mechanisms for gene transcription.

## 5. Conclusions

In conclusion, we demonstrated for the first time that the ORF of the ASFV *B438L* gene was governed by two different promoters, and the distal promoter could also initiate the transcription of both the *B438L* and *B169L-2* genes (Figure 6). A comprehensive understanding of the multiple promoters of viral genes can facilitate the development of more effective antiviral strategies.

## Figures and Tables

**Figure 1 viruses-16-01058-f001:**
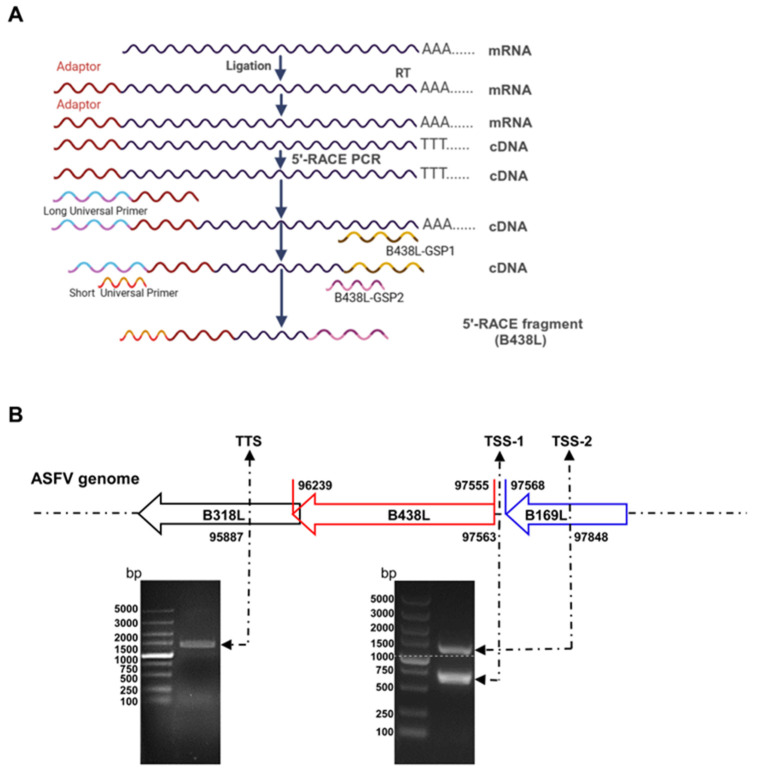
Two distinct initiation sites for the transcription of the *B438L* gene of African swine fever virus. (**A**) Preparation of cDNAs. Total RNA was obtained from the ASFV-infected PAMs at 24 h postinfection, and cDNAs were synthesized using the SMARTer RACE 5′/3′ kit. After removal of the 5′ phosphate groups and the 5′ cap structure using calf intestine alkaline phosphatase and tumor abnormal protein, respectively, the adaptor was fused to the 5′-terminus of the mRNA using T4 RNA ligase. RT, reverse transcription. (**B**) Identification of the *B438L* transcript. At the upper panel, the numbers represent the respective loci of the initiation codon ATG and the stop codon TAG of the *B438L* gene in the genome of the ASFV HLJ/2018 strain. The bottom panel shows that the PCR products were run in a 1.2% agarose gel. The numbers on either side of the image indicate the corresponding sites of TSS-1 (lower band), TSS-2 (upper band), and the TTS in the ASFV genome. TSS, transcription start site; TTS, transcription termination site.

**Figure 2 viruses-16-01058-f002:**
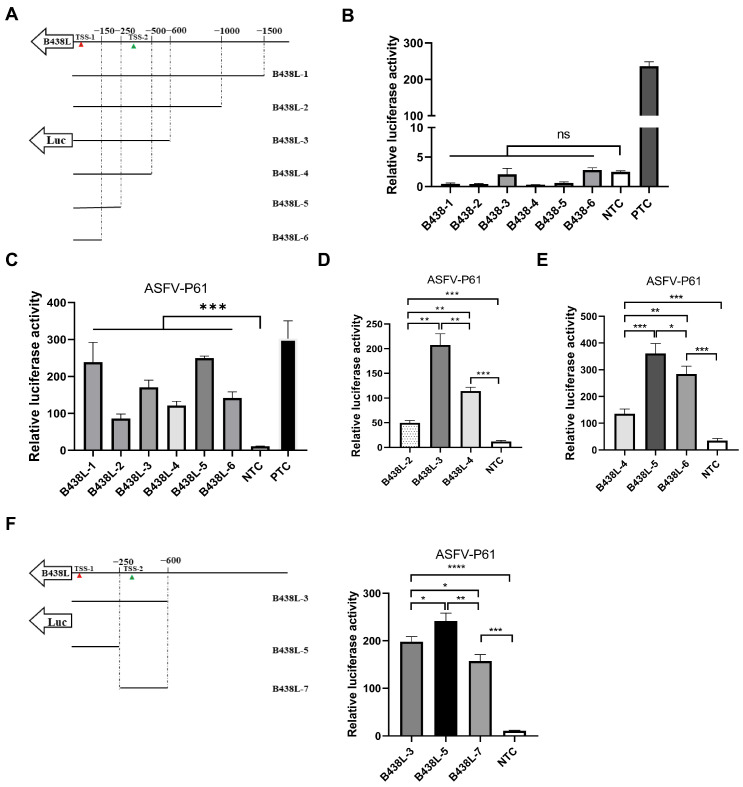
Mapping of the *B438L* promoter in the *B169L* gene. (**A**) Schematic diagram of the *B438L* promoters. Each DNA fragment was obtained by PCR using the primers listed in Table 1. (**B**) None of the promoters of *B438L* exhibited promoter activities. HEK293T cells in 6-well plates were transfected with the reporter plasmids (2 μg/each) of *B438L-1* to *-6* for 36 h, respectively. Subsequently, the cells were lysed, and the activities of the *B438L* promoters were analyzed by luciferase reporter assay. (**C**) *B438L-1*, *-3*, and *-5* exhibited robust promoter activities upon ASFV-P61 infection. HEK293T cells were transfected with the reporter plasmids as described above. Then, the cells were infected with an ASFV-P61 strain at a multiplicity of infection of 0.1 for 24 h. The cells were lysed and subjected to the examination of firefly luciferase and *Renilla* luciferase activities. The data represent the firefly luciferase activity, normalized to that of the *Renilla* luciferase. (**D** and E) The comparison of promoter activities among *B438L-2*, *-3*, and *-4* (**D**) or *B438L-4*, *-5*, and *-6* (**E**). (**F**) The identification of the promoter activity of *B438L-7*. The schematic diagram of the promoters of *B438L-3*, *-5*, and *-7* is shown in the left panel. The promoter activities of *B438L-3*, *-5*, and *-7* are shown in the right panel. All the experiments were performed in triplicates. NTC, negative control, pGL3-Basic; PTC, positive control, pGL3-Control. All the data were analyzed using the Student’s *t* test. *, *p* < 0.05; ** *p* < 0.01; *** *p* < 0.001; **** *p* < 0.0001, *ns*, not significant.

**Figure 3 viruses-16-01058-f003:**
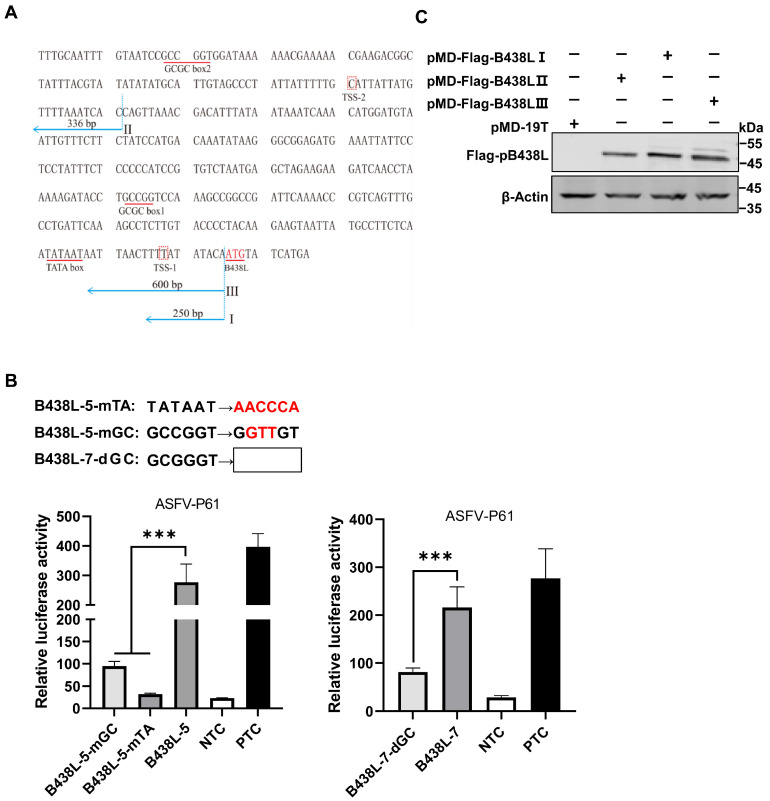
Identification of the TATA and GC boxes in the *B438L* promoter. (**A**) The predication of TATA and GC boxes in the *B438L* promoter. The TATA and GC boxes were predicted and analyzed by GPMiner (https://bio.tools/GPMiner (accessed on 13 June 2023)), JASPAR (https://jaspar.genereg.net (accessed on 14 June 2023)), and Softberry (https://www.softberry.com (accessed on 16 June 2023)). The sequences shown in Figure 3A represent the reverse-complemented DNA sequences of the promoter regions. The TSSs were examined by 5′RACE and indicated as TSS-1, -2, respectively. (**B**) The identification of the functional TATA and GC boxes in the *B438L* promoter. HEK293T cells in 24-well plates were transfected with the reporter plasmids (2 μg/each) with the mutation or deletion of TATA and the GC box of the *B438L* promoter, and the cells were infected with the ASFV-P61 strain at an MOI of 0.1 for 24 h. The activities of the *B438L* promoters were determined by luciferase reporter assay. The data represent the firefly luciferase activity, normalized to the *Renilla* luciferase activity. The experiments were carried out in triplicates. (**C**) Both promoter regions (*B438L*-5, -7) initiate the expression of *B438L*. The plasmids harboring *B438L* under the control of various promoter sequences (2 μg/each), i.e., pMD-Flag-B438L-I (*B438L-5*, the region of nt 97555–97804), -II (*B438L-7*, the region of nt 97819–98154), or -III (*B438L-3*, the region of nt 97555–98154) or the empty vector pMD-19T, were transfected into HEK293T cells for 24 h. Then, the cells were infected with the ASFV-P61 strain as described above. Subsequently, the immunoblotting analyses were performed with an anti-Flag monoclonal antibody. NTC, negative control, pGL3-Basic; PTC, positive control, pGL3-Control. All the data were analyzed using the Student’s *t* test. *** *p* < 0.001.

**Figure 4 viruses-16-01058-f004:**
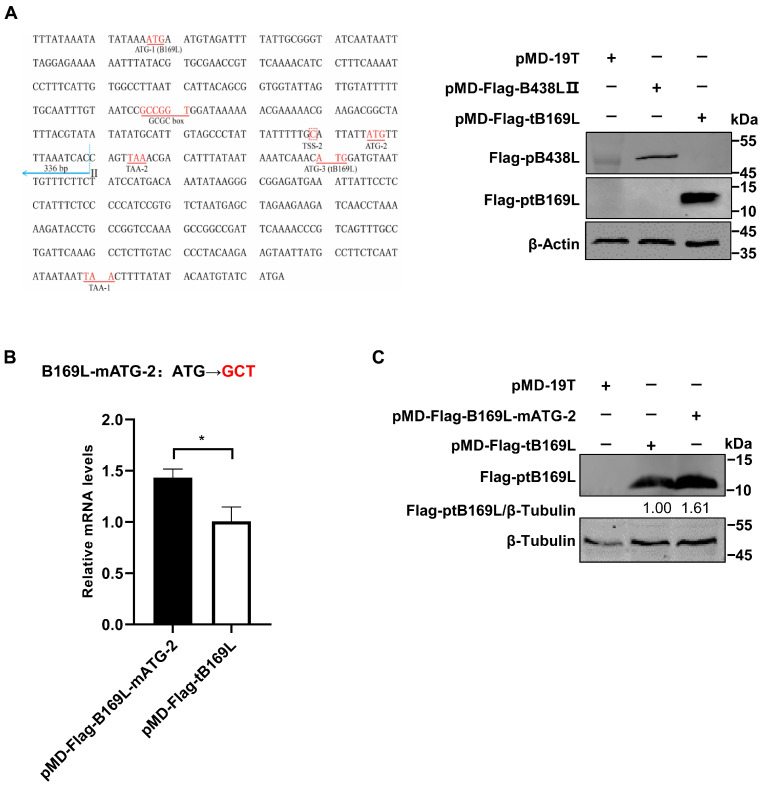
Identification of the *B438L* promoter that initiates the transcription of *B169L-2*. (**A**) The *B438L* distal promoter is capable of initiating the transcription of *B169L-2*. HEK293T cells in 6-well plates were transfected with pMD-Flag-tB169L or pMD-Flag-B438L-II (2 μg/each), followed by infection with the ASFV-P61 strain as described above. Subsequently, the immunoblotting analysis was performed with an anti-Flag monoclonal antibody (MAb). The sequences shown in Figure 4A (left panel) represent the reverse-complementary DNA sequences of the promoter regions. (**B**,**C**) The *B169L-2* transcription efficiency is affected by the second initiation codon (ATG-2). HEK293T cells were transfected with pMD-Flag-tB169L or pMD-Flag-B169L-mATG-2 (2 μg/each) and infected with the ASFV-P61 strain as described above. Subsequently, the transcription level of *B169L-2* was measured by relative reverse transcription–quantitative PCR (**B**), and the protein expression of ptB169L (encoding by *B169L*-mRNA2) was analyzed by immunoblotting analysis using the anti-Flag MAb (**C**). All the data were analyzed using the Student’s *t* test. *, *p* < 0.05.

**Figure 5 viruses-16-01058-f005:**
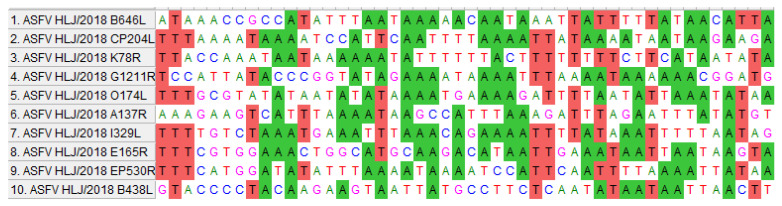
Alignment of 5′-flanking sequences of the ASFV genes. The selected 50-bp sequences from the upstream of transcription start site were aligned to compare the nucleotide sequence identities.

**Figure 6 viruses-16-01058-f006:**
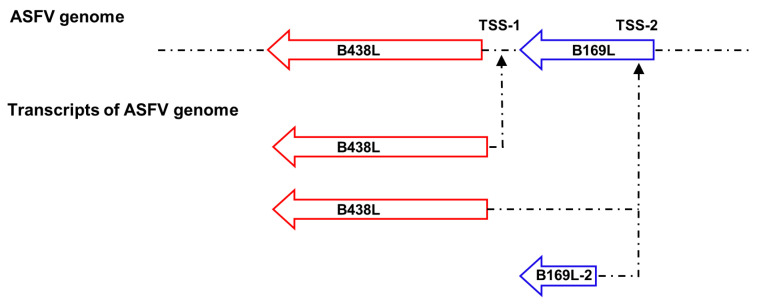
A working model for the transcription of the *B438L* and *B169L-2* genes of ASFV. The *B438L* gene is governed by two different promoters, and the distal promoter also initiates the transcription of both the *B438L* and *B169L-2* genes.

**Table 1 viruses-16-01058-t001:** The sequences of the primers used for PCR.

Primers	Sequences (5′–3′)	Description
Long Universal Primer	CTAATACGACTCACTATAGGGCAAGCAGTGGTATCAACGCAGAGT	B438L
Short Universal Primer	CTAATACGACTCACTATAGGGC	B438L
GSP1	GATTACGCCAAGCTTAGCATCGGCACGTCCGTGTAGGTAC	B438L
GSP2	GATTACGCCAAGCTTGTGTGCTCGCTGAACCTCGTAGAAG	B438L
B438L-1.F	CCGCTCGAGTGTATATAAAAGTTAATTATTATATTG	
B438L-1.R	GGGGTACCTGCGACTACATTAGCAATCTGGGCACC	B438L-1
B438L-2.F	GGGGTACCCCATTAGATAACTATCCCGTGCCAC	B438L-2
B438L-3.F	GGGGTACCAGTCGGGCCTTTTTGAAGAATCTTC	B438L-3
B438L-4.F	GGGGTACCTAGATTTTATTGCGGGTATCAATAA	B438L-4
B438L-5.F	GGGGTACCGCCCTATTATTTTTGCATTATTATG	B438L-5
B438L-6.F	GGGGTACCATAAGGGCGGAGATGAAATTATTCC	B438L-6
B438L-7.F	GGGGTACCTGCGACTACATTAGCAATCTGGGCACC	B438L-7
B438L-7.R	CCGCTCGAGTACAATGCATATATATACGTAAATAGC	
B438L-5-mTA.F	CCTTCTCAAAACCAAAATAATTAAC	B438L-5-mTA
B438L-5-mTA.R	GTTAATTATTTTGGTTTTGAGAAGG	
B438L-5-mGC.F	ATTATGCCTTCTCAAAATTAACTTT	B438L-5-mGC
B438L-5-mGC.R	AAAGTTAATTTTGAGAAGGCATAAT	
B438L-7-dGC.F	AATGTAGATTTTATTATCAATAATTTAGGA	B438L-7-dGC.
B438L-7-d.GC.R	TCCTAAATTATTGATAATAAAATCTACATT	
pMD-Flag-B438L-I.F	CGGATCCTATAATAAATCAAACATGGATGTA	pMD-Flag-B438L-I
pMD-Flag-B438L-I.R	TAAGAACTACTTATCGTCGTCATCCTTGTAATCCAATGATGGAGATATAGATG	
pMD-Flag-B438L-II.1F	CGGATCCGTCGGGCCTTTTTGAAGAATCTTCA	pMD-Flag-B438L-II
pMD-Flag-B438L-II.1R	CATAATCATGATACATTGATTTAAAACATAA	
pMD-Flag-B438L-II.2F	TTATGTTTTAAATCAATGTATCATGATTATG	
pMD-Flag-B438L-II.2R	TAAGAACTACTTATCGTCGTCATCCTTGTAATCCAATGATGGAGATATAGATG	
pMD-Flag-B438L-III.F	CGGATCCGTCGGGCCTTTTTGAAGAATCTTCA	pMD-Flag-B438L-III
pMD-Flag-B438L-III.R	TAAGAACTACTTATCGTCGTCATCCTTGTAATCCAATGATGGAGATATAGATG	
pMD-Flag-tB169L.F	CGGATCCTCCTTTCAAAATCCTTTCATTGTGGC	pMD-Flag-tB169L
pMD-Flag-tB169L.R	GGAATTCTTACTTGTCGTCATCGTCTTTGTAGTCATTATTATATTGAGAAGGC	
pMD-Flag-B169-mATG-2.F	TATTATTTTTGCATTATTGCTTTTTAAATCACCAGTTAA	pMD-Flag-B169-mATG-2
pMD-Flag-tB169-mATG-2.R	TTAACTGGTGATTTAAAACATAATAATGCAAAAATAATA	
qB169L-2-F	TTTGCAATTTGTAATCCGCCGGTG	RT-qPCR for *B169L*-mRNA2
qB169L-2-R	ACTTCTTGTAGGGGTACAAGAGG	

## Data Availability

Data supporting the reported results are available in this article.

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
