# Peer review of "The Distal Promoter of the B438L Gene of African Swine Fever Virus Is Responsible for the Transcription of the Alternatively Spliced B169L"

_viruses, 2024, doi:10.3390/v16071058_

Round 1

Reviewer 1 Report

Comments and Suggestions for Authors

It is true that ASFV is a highly significant pathogen in swine and wild boar worldwide. However, the purpose of the author's study on the B438L gene of this virus is unclear. It appears to be research conducted for the sake of research itself.

The introduction should clearly explain why this gene is important and why the study was conducted on it.

There is no explanation in the materials and methods section of what HVRI stands for.

- Explain why primary PAM cells and HEK cells were used.

- The description of materials lacks the company, city, and country details.

In the discussion section, instead of a deep reflection on their research, the authors merely refer to similar studies. Citing references stating that CMV, EBV, etc., have multiple promoters and them concluding that the gene in ASFV under study also has multiple promoters does not seem to align with the seesnce of what a research paper's discussion should pursue.

Author Response

It is true that ASFV is a highly significant pathogen in swine and wild boar worldwide. However, the purpose of the author's study on the B438L gene of this virus is unclear. It appears to be research conducted for the sake of research itself.

Author’s response: Thank you for the professional suggestions. The B438L gene is located adjacently in the upstream B169L gene (with a 14-bp interval sequence) in the ASFV genome. Therefore, we presumed the transcriptional association between the B438L and B169L genes. The purpose of this study is to identify the promoter(s) within the B169L gene to facilitate the generation of the B169L-deleted ASFV mutant, which has been indicated in the Introduction (Lines 40–47 and 67–69).

  1. The introduction should clearly explain why this gene is important and why the study was conducted on it.

Author’s response: B438L and B169L genes, both of which encode structural proteins of ASFV, are arranged adjacently in the ASFV genome. It has been shown that B438L is indispensable for ASFV replication. However, the functional role of B169L during ASFV infection remains unknown. To construct the B169L-deleted ASFV mutant without impairing the promoter activities of B438L, it is crucial to identify the promoter region within the B169L gene (Lines 44–47 and 67–69).

  1. There is no explanation in the materials and methods section of what HVRI stands for.

Author’s response: HVRI, the abbreviation of Harbin Veterinary Research Institute, has been included in the manuscript (Lines 85–86).

  1. Explain why primary PAM cells and HEK cells were used.

Author’s response: PAMs, the target cells of ASFV, were used for viral infection and subsequent viral genome quantification (Lines 96–97), while HEK293T cells were used to assess promoter activities (Lines 124–131).

  1. The description of materials lacks the company, city, and country details.

Author’s response: The detailed information of the materials has been included in the revised manuscript (Lines 89–92, 98–99, 103, 107,115–117,119,133–134,141,143).

  1. In the discussion section, instead of a deep reflection on their research, the authors merely refer to similar studies. Citing references stating that CMV, EBV, etc., have multiple promoters and them concluding that the gene in ASFV under study also has multiple promoters does not seem to align with the sequence of what a research paper's discussion should pursue.

Author’s response: The transcription phases of the ASFV genes are complex and remain to be elucidated. In this study, we reported a novel strategy, which enables the identification of the viral gene promoters through the transfection with the promoter reporter in combination with the infection with the HEK293T cells-adapted ASFV and revealed the transcription association between the B438L and B169L genes. Furthermore, we have demonstrated that the transcription of several viral genes of ASFV depends on the viral RNA polymerases (Lines 329–337, 343–351, 358–366, and 371–380).

Reviewer 2 Report

Comments and Suggestions for Authors

In comparison to their cellular counterparts, the viral transcription systems are poorly understood, despite the components of the viral transcription system can be considered as therapeutic targets in the treatment of viral disease. Viruses use multiple strategies to regulate gene expression. For example, several viral ORFs may be under the control of the same promoter. The alternatively spliced ​​mRNAs have been described for many viruses of different families. Such alternative ORFs often encode proteins that modulate the host immune response. Therefore, this work, demonstrating that the ORF of the ASFV B438L gene was governed by two different promoters, and the distal promoter could also initiate the transcription of both B438L and B169L-2 genes, is interesting and important for studying the ASFV transcription mechanisms. The article is well written, and the methodology used is adequate. The authors have experience in studying animal viruses, including the ASF virus.

The authors should add information about previously characterized ASFV promoters and conduct a small comparative analysis. This information should be added to the Discussion section to improve the manuscript.

Author Response

In comparison to their cellular counterparts, the viral transcription systems are poorly understood, despite the components of the viral transcription system can be considered as therapeutic targets in the treatment of viral disease. Viruses use multiple strategies to regulate gene expression. For example, several viral ORFs may be under the control of the same promoter. The alternatively spliced ​​mRNAs have been described for many viruses of different families. Such alternative ORFs often encode proteins that modulate the host immune response. Therefore, this work, demonstrating that the ORF of the ASFV B438L gene was governed by two different promoters, and the distal promoter could also initiate the transcription of both B438L and B169L-2 genes, is interesting and important for studying the ASFV transcription mechanisms. The article is well written, and the methodology used is adequate. The authors have experience in studying animal viruses, including the ASF virus.

Author’s response: We greatly appreciate the constructive suggestions and positive comments.

  1. The authors should add information about previously characterized ASFV promoters and conduct a small comparative analysis. This information should be added to the Discussion section to improve the manuscript.

Author’s response: As suggested, we have conducted a multiple sequence comparison among the previously characterized promoters of ASFV, including the B438L promoter. Interestingly, the promoters exhibit a significant enrichment in A and T (the updated Figure 5) (Lines 393–403).

Round 2

Reviewer 2 Report

Comments and Suggestions for Authors

The manuscript is interesting and important for studying the ASFV transcription mechanisms. The article is well written, and the changes made by the authors in the Discussion section have improved the manuscript. The manuscript is worthy of publication.